



# Effect of chemically induced fracturing on the ice nucleation activity of alkali feldspar.

Alexei Kiselev[1], Alice Keinert[1], Tilia Gaedeke[1], Thomas Leisner[1,2], Christoph Sutter[3], Elena Petrishcheva[3], Rainer Abart[3]

[1] Karlsruhe Institute of Technology, Institute of Meteorology and Climate Research, Karlsruhe, Germany
[2] Institut für Umweltphysik, Universität Heidelberg, Heidelberg, Germany
[3] University of Vienna, Department of Lithospheric Research, Althanstrasse 14, A-1090 Vienna, Austria

*Correspondence to*: Alexei Kiselev (alexei.kiselev@kit.edu)

**Abstract.** Feldspar is an important constituent of airborne mineral dust. Some alkali feldspars exhibit particularly high ice nucleation (IN) activity. This has been related to structural similarities of the ice (10-10) prism planes and the (100) planes of alkali feldspar. Here the effect of generating surfaces with close to (100) orientation by means of chemically induced fracturing on the IN activity of alkali feldspar was investigated experimentally. Gem quality K-rich alkali feldspar was shifted towards more Na-rich compositions by cation exchange with an NaCl-KCl salt melt at 850°C, and a system of parallel cracks with an orientation close to (100) was induced. Droplet freezing assay experiments performed on grain mounts of the cation exchanged alkali feldspars revealed an increase of the overall density of ice nucleating active sites (INAS) with respect to the untreated feldspar. In addition, annealing at 550°C subsequent to primary cation exchange further enhanced the INAS density and lead to IN activity at exceptionally high temperatures. Although very efficient in experiment, fracturing by cation exchange is unlikely to be of relevance in the conditioning of alkali feldspars in nature. However, parting planes with similar orientation as the chemically induced cracks may be generated in lamellar microstructures resulting from the exsolution of initially homogeneous alkali feldspar, a widespread phenomenon in natural alkali feldspar known as perthite formation. Perthitic alkali feldspars indeed show the highest IN activity. We ascribe this phenomenon to the preferential exposure of crystal surfaces oriented sub-parallel to (100).

## 1 Introduction - mineralogical aspects and relevance for atmospheric science

Feldspar is the most abundant mineral in the Earth's crust. It is a major constituent of magmatic, metamorphic, and sedimentary rocks (Smith and Brown, 1988), and due to its ubiquity on the Earths' surface, feldspar is also an abundant constituent of the solid aerosol particles. Desert dust (Boucher et al., 2014) and volcanic ash (Durant et al., 2008; Durant et al., 2010) are the main sources of airborne mineral dust contributing about 1000 – 4000 Tg/a and 176 – 256 Tg/a, respectively. Solid aerosol particles are of interest in the context of ice formation in clouds. Mineral dust may substantially increase the freezing temperature of supercooled cloud droplets and foster the formation of ice particles at relatively high temperatures (Tang et al., 2016). Atmospheric ice particles have a strong influence on the physical properties of clouds and exert first order controls on processes such as radiative transfer, precipitation and absorption of trace gases (Kanji et al., 2017). The albedo of clouds generally increases with the formation of ice particles (McFarquhar et al., 2002), an effect that is of pivotal importance for the Earth's radiation budget und is thus an important factor for global climate (Boucher et al., 2014; Bony et al., 2006). Among the different types of airborne mineral dust particles some feldspars have been reported to have particularly high ice nucleation activity (Atkinson et al., 2013; Harrison et al., 2016; Peckhaus et al., 2016).

Feldspar is a framework silicate and forms a ternary solid-solution among the Ca- ($Ca_2Al_2Si_2O_8$ - anorthite), Na ($NaAlSi_3O_8$-albite), and K ($KAlSi_3O_8$-K-feldspar) end-members, where the latter two pertain to the alkali feldspar solid-solution. The crystal structure of feldspar is comprised of a three-dimensional framework of corner-sharing $SiO_4$ and $AlO_4$ tetrahedrons with the alkali- and alkali earth cations located in large framework cavities. Depending on chemical composition, pressure,



temperature, and the state of Al-Si ordering on the tetrahedral sublattice, feldspar may have monoclinic C2/m or triclinic C1 symmetry (Ribbe, 1983). The mineralogy of feldspar has a crucial effect on its ability to cause freezing of supercooled water. The ice nucleating efficacy expressed in terms of the surface density of ice nucleating active sites (INAS, $n_s(T)$), (Connolly et al., 2009) was shown to vary over two orders of magnitude for various alkali feldspars (Harrison et al., 2016). Perthites, which are exsolved feldspars that typically take the form of (sub) micron scale lamellar intergrowth of more K-rich and more

Na-rich alkali feldspar, were found to have the highest IN efficacy among all feldspar dust particles (Whale et al., 2017). The mechanisms relating the lamellar microstructure formed by exsolution and enhanced IN efficacy are, however, not known and are currently debated.

In a previous study that used natural perthitic alkali feldspars, the preferential epitaxial nucleation and growth of ice crystals by the alignment of the ice (10-10) prism plane with (100) faces of the feldspar was identified as the key mechanism underlying

the high IN activity of alkali feldspar in deposition freezing (Kiselev et al., 2017). Facets with (100) orientation, however, do not pertain to the commonly exposed crystal surfaces of alkali feldspar, which typically has (110), (-101), (001), (20-1), and (010) growth facets (Smith and Brown, 1988). In addition, (001) and (010) surfaces are frequently exposed on the surfaces of natural alkali feldspar due to excellent cleavage parallel to these planes (Smith and Brown, 1988). Crystal surfaces with (100) orientation have high surface energy, and (100) facets or cleavage planes hardly ever occur. It has been argued that small

patches of (100) crystal surfaces may be exposed at defects such as cracks and cavities (Fitz Gerald et al., 2006; Kiselev et al., 2017; Whale et al., 2017). In this context perthites are of interest. The boundaries between the more Na-rich and the more K-rich lamellae of perthite are oriented so that the crystallographic lattice misfit between the compositionally distinct lamellae is minimized. This condition defines the so called *Murchison plane* (Smith and Brown, 1988), which is a plane with non-rational Miller indices between (-601) and (-801), that is, a plane that does not coincide with any of the primary atomic planes. The

orientation of the *Murchison plane* is within 8° to 11° off from (100), and parting planes following the Murchison plane may well contain patches of (100) crystal surface. In the light of the extraordinarily high IN activity of alkali feldspar and the preferential epitaxial nucleation of ice on feldspar (100) surfaces, it is important to understand the mechanisms by which such surfaces may form and to what extent the IN activity of these surfaces differs from the IN activity of the more commonly exposed growth facets and cleavage planes. In particular, the potential effects of different formation mechanisms and

associated crystal surface morphologies on the efficacy of IN on alkali feldspar aerosol particles is of interest. In this communication we investigate the IN activities of alkali feldspars that were subject to different pre-treatments designed to mimic natural processes leading to the exposure of crystal surfaces sub-parallel to (100). We relate the IN activity to the mode and the extent of specimen alteration and discuss the potential role of (100) surfaces in the IN activity of alkali feldspars.

## 2 Experiment

### 2.1 Sample preparation

### 2.1.1 Cation exchange and annealing experiments

For preparing feldspar fragments with a significant fraction of (100) surfaces we made use of the fact that fracturing of alkali feldspar sub-parallel to the Murchison plane may be induced by shifting the composition of a K-rich alkali feldspar towards more Na-rich compositions by diffusion mediated Na-K cation exchange with an NaCl-KCl salt melt (Neusser et al., 2012;

Petrishcheva et al., 2019; Predan et al., 2020; Scheidl et al., 2014). This effect is due to the strongly anisotropic contraction of the crystal structure with increasing Na content (Kroll et al., 1986; Angel et al., 2012) with the strongest contraction approximately perpendicular to the Murchison plane. During diffusion mediated cation exchange with NaCl-KCl salt melt of appropriate composition a more Na-rich layer forms on the surface of the specimen, while the original more K-rich composition is retained in the internal regions. Both, the internal regions with zero compositional eigenstrain state and the more Na-rich

surface layer with strong negative eigenstrain state perpendicular to the Murchison plane pertain to one solid and a tensile



stress state is induced in the chemically altered surface layer. When the compositional shift towards more Na-rich compositions of the surface layer exceeds a certain extent, a critical stress level is reached, followed by fracturing approximately parallel to the Murchison plane that is perpendicular to the direction of strongest contraction (Neusser et al., 2012; Petrishcheva et al., 2019; Scheidl et al., 2014; Predan et al., 2020).

We exploited this mechanical effect for producing surfaces with orientations close to (100) by performing cation exchange experiments using powders of gem-quality alkali feldspar and a NaCl-KCl salt mixture as the starting materials. The original feldspar is a sanidine from Volkesfeld (Eifel, Germany) with $c_K^{fsp} = 0.84$, where $c_K^{fsp}$ is the atomic site fraction of K on the alkali sublattice, $c_K^{fsp} = [K]/([Na]+[K])$ in atomic units (marked with the red solid circle in the phase diagram in the Figure 1). The feldspar was gently crushed and sieved, and the grains of the 100-200 µm sieve fraction were exchanged with a mixed

molten NaCl-KCl salt with a composition of $c_K^{salt} = 0.21$ at 850°C for 8 days to attain equilibrium Na/K partitioning between feldspar and the salt. The salt was applied in excess so that it practically retained its composition unchanged during the cation exchange experiment. At 850°C the composition of Volkesfeld sanidine in equilibrium with NaCl-KCl salt melt with $c_K^{salt} = 0.21$ is $c_K^{fsp} = 0.43$ (Neusser et al., 2012). The induced compositional change corresponds to a shift of $\Delta c = 0.41$ towards more Na-rich compositions. This shift is sufficiently large to induce fracturing, and a system of parallel cracks with an orientation

close to the Murchison plane developed in the cation exchanged feldspars.

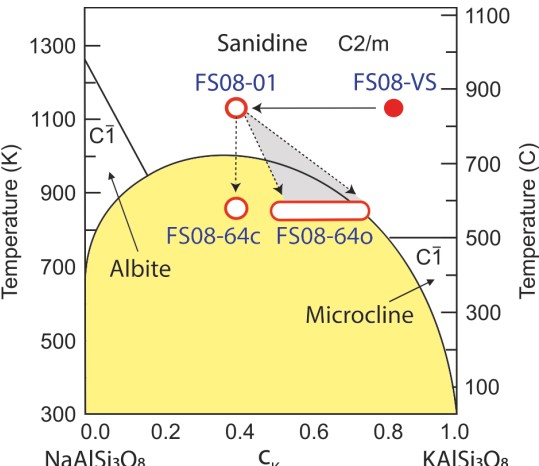

**Figure 1: Schematic isobaric phase diagram for alkali feldspar showing the stability fields and respective symmetries of albite,**
**sanidine and microcline; the miscibility gap (the yellow-shaded region under the solvus curve) represents the two-phase region; the filled red circle shows the original composition of the Volkesfeld sanidine (sample FS08-VS), which was shifted to intermediate composition by cation exchange at 850°C (sample FS08-01); subsequent annealing at 550°C in vacuum produced sample FS08-64c, and annealing at 550°C in contact with the NaCl-KCl salt produced sample FS08-64o with a somewhat variable composition due to incomplete cation exchange (see below).**

After the cation exchange, the feldspar samples were split into three batches. One batch was annealed at 550°C for 64 days, while the grains were kept in contact with the salt so that further cation exchange between the salt and the feldspar occurred. At 550°C the salt is solidified and cation exchange is slower than during the primary cation exchange, but it is still sufficiently fast so that the resulting chemical alteration can well be detected (see below). We refer to this annealing route as the *open system setting*, and the sample produced by this route is labelled FS08-64o. A second batch of cation exchanged feldspar grains

was removed from the salt and annealed at 550°C in vacuum for 64 days. The conditions lay within the miscibility gap of alkali feldspar (the yellow-shaded region in Figure 1), that is a region on the isobaric phase diagram, where two alkali feldspars of different composition coexist in equilibrium (Work et al., 2004; Brown and Parsons, 1984) and sub-micron scale phase separation of the originally compositionally homogeneous feldspar with $c_K^{fsp} = 0.43$ into more K- and more Na-rich feldspars





occurred (Petrishcheva et al., 2020). We refer to this annealing route as the *closed system setting*, and the sample produced is labelled FS08-64c. A third batch of cation exchanged feldspar grains was removed from the melt after the primary cation exchange and stored at room temperature without further treatment. This sample is labelled FS08-01. The details of the treatment are illustrated in Figure 1 and the properties of the resulting feldspar grains are summarized in Table 1. Grain mounts were prepared from all three sample batches as described in the next section.

### 2.1.2 Preparation of grain mounts and thin sections

The amount of chemically altered feldspar prepared as described in the previous section was too small for conducting droplet freezing experiments with aqueous suspensions of feldspar powder, as usually done in mineral dust IN efficiency studies (as in, e.g. Hiranuma et al. (2015). As an alternative, we prepared grain mounts by casting feldspar grains in a thin layer of epoxy resin on top of a standard microscope glass slide. After solidification of the resin the layer of cast epoxy-feldspar aggregate was ground to a thickness of 200 µm and subsequently polished to optical quality. Through this procedure, the majority of the

feldspar grains was exposed on the polished sample surface. The grain mounts were cut into 10x10 mm square plates and end-polished using a Leica TIC3X ion-milling device. A current of 0.5 mA and an acceleration voltage of 1 kV was applied for 45 min, and the sample was continuously rotated in the horizontal plane. The same procedure was applied for preparing grain-free mounts of epoxy resin (FS08-Epx) to be used for reference measurements.

The grain mounts were analyzed by polarization microscopy using a Leica DM4 polarization microscope and in a FEI Thermo

Fisher Quattro Environmental Scanning Electron Microscope (ESEM) equipped with an EDAX Octane Super energy dispersive X-Ray spectrometer (EDS). Exemplary views of the grain mount prepared from the cation exchanged alkali feldspar (sample FS08-01) are shown in Figure 2.

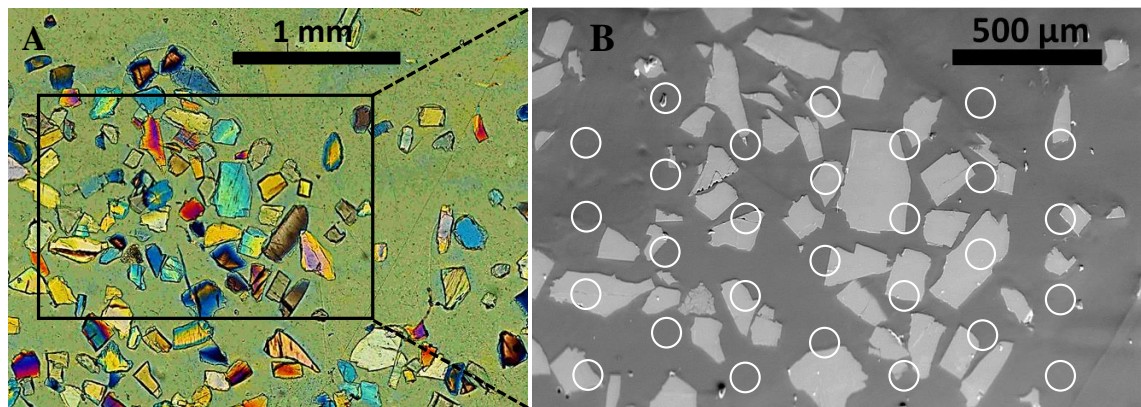


**Figure 2. Grain mounts used in the droplet freezing assay study. A. Optical microscope image in transmitted light with crossed-polarizers and first-order retardation plate showing feldspar grains embedded in epoxy resin (sample FS08-01). B. Back-Scattered Electron (BSE) image of the area marked by the black frame in panel A. The white circles illustrate the footprints of water droplets used in the cold stage experiment.**

For comparison, a natural perthitic alkali feldspar from Pakistan (sample FS06-010) was analysed. The sample was prepared from a centimetre-sized specimen with partially developed facets. Visually, the feldspar specimen is pale beige and turbid (see Supplementary Figure 1). The X-Ray diffraction (XRD, Panalytical, Cu K-alpha 1&2) analysis yielded a phase content of 41% orthoclase, 39% microcline, and 20% albite (see supplemental material for details). In transmitted light optical microscopy and in the SEM a complex microperthitic structure overlain by polysynthetic twinning after the pericline law and porosity,

which is very likely related to hydrothermal or deuteric alteration, were identified (see Supplementary Figure 2). For our experiments the specimen was cut parallel to the (010) cleavage plane, and a thin section was prepared employing the same


polishing routine as used for the grain mounts. The crystallographic orientation of the cleavage plane was confirmed by backscattered electron diffraction (BSED) obtained with EDAX Hikari Super BSED detector mounted on the ESEM. The BSED patterns were analysed with the EDAX OIM v8.0 software package using the built-in structure file of generic triclinic

feldspar.

**2.2 Droplet freezing assay experiments**

**2.2.1 Measurement routine and calculation of INAS density**

The ice nucleating efficacy of the altered feldspar was measured in the droplet freezing assay setup previously described in (Peckhaus et al., 2016). Briefly, a 10-by-10 mm grain mount was placed into a temperature-controlled cold stage setup

consisting of a Linkam MDBCS-196 motorized cold stage, a piezo-driven drop-on-demand dispenser (GeSIM, model A010-006 SPIP) and a video camera with a wide-field objective allowing for detection of individual freezing events with 0.125 s time resolution. In the experiments described in this work, nearly 500 droplets of pure water each with a volume of approximately 0.4 nL were deposited onto the polished surface of the grain mount in a chequerboard pattern with 400 µm center-to-center separation distance between the droplets (see Figure 2B). During the freezing experiments the cold stage was

cooled with a rate of 3 K/min. The temperature was monitored with a thin-film platinum resistance sensor (Pt-100) that was fixed directly on the sample surface using a vacuum grade heat-conducting paste. Freezing of individual droplets was detected by an intensity change of the light reflected by the droplets, which happens at the moment of freezing and is best detected using crossed polarizers. An automated LabView video analysis routine was used to identify the individual droplet positions and freezing temperatures and for calculating the fraction of frozen droplets as a function of temperature $f_{ice}(T)$ (see Figure

5A). To account for the different freezing efficacies of feldspar and epoxy resin, several freezing experiments were conducted with the droplets deposited on the surface of a feldspar-free epoxy mount (sample FS08-Epx). For each sample, at least three replicate measurements were done. Between successive measurements the droplets were evaporated and redeposited in the similar pattern. Note, however, that due to the limited positioning accuracy of the motorized cooling stage, the new pattern could be shifted with respect to the previous one by a distance of up to 100 µm in random direction. During the cooling ramp,

some droplets located near the side of the grain mount have reduced their sizes or disappeared completely due to evaporation. These droplets were excluded from the post-processing analysis, resulting in a fewer total number of droplets actually used for evaluation of the INAS density of feldspar (see Table 1 for the actual total number of droplets in every experiment). The droplet footprint area was obtained by measuring 20 randomly chosen droplet contours in a video frame recorded just before the detection of the first freezing event. In this way the gradual droplet evaporation and associated reduction of footprint area

during the cooling ramp could be taken into account.

The heterogeneous freezing efficacy of a substrate is usually expressed in terms of the INAS density (Murray et al., 2012; Connolly et al., 2009). Within the framework of the classical nucleation theory (CNT, see e.g. (Hoose and Möhler, 2012), the INAS density $n_s(T)$ at temperature $T$ is obtained from the fraction of frozen droplets $f_{ice}(T)$ and the area of contact $S_d$ between the droplet and the substrate:

$$n_s(T) = -\frac{\ln(1 - f_{ice}(T))}{S_d} \quad\quad\quad\quad\quad (1)$$

The average water-feldspar contact area was determined from the droplet size and from the exposed surface area of the feldspar grains. Due to the comparable size of the footprints of the droplets and the feldspar fragments exposed on the sample surface, a droplet can be in contact with feldspar and epoxy resin or it may be in contact with epoxy resin exclusively (see Figure 2B). This is why the IN efficacy of the epoxy resin needs to be considered. Moreover, in the replicate experiments the droplets were

deposited on slightly different positions, and therefore the water-feldspar contact area did not only vary from droplet to droplet but also between replicate measurements. The water-feldspar contact area was evaluated using the back-scattered electron microscope images of the grain mounts (Figure 3). A black-and-white binary mask created from the segmented image of a




grain map was overlain with the contour image of a droplet array, as illustrated in Figure 3, and the average overlapping area $S_{FS}$ per droplet was determined. Given that the number of IN active sites associated with the surface of feldspar grains $n_s^{FS}(T)$

and the number of INAS associated with the surface of epoxy resin $n_s^{EPX}(T)$ are additive, we may write:

$$n_s^{app}(T) \cdot S_d = n_s^{FS}(T) \cdot S_{FS} + n_s^{EPX}(T) \cdot S_{EPX} \qquad (2)$$

where $n_s^{app}(T)$ is the apparent INAS density obtained from the droplet freezing experiments via equation 1, and $S_{FS}$ and $S_{EPX}$ are the average droplet-feldspar and droplet-epoxy contact areas, which satisfy the condition $S_d = S_{FS} + S_{EPX}$. Combining equations 1 and 2, the INAS density associated with feldspar grains is calculated as

$$n_s^{FS}(T) = -\frac{\ln(1-f_{ice}(T))}{S_{FS}} - n_s^{EPX}(T) \cdot \left(\frac{S_d}{S_{FS}} - 1\right), \qquad (3)$$

with $n_s^{EPX}(T)$ obtained from the droplet freezing experiments conducted on the grain-free mount of epoxy resin (sample FS08-Epx).

The actual position of every droplet could not be measured precisely as the resolution of the main camera was not sufficient to recognize the footprint of every droplet automatically. Therefore, the droplet array was created using the coordinates of the

individual droplets identified by the video processing software and the diameter of droplet footprint measured manually for several droplets clearly visible in the video frames. To estimate the error arising from use of such *synthetic* droplet array, the calculation of $S_{FS}$ was repeated 100 times for every grain mount with the position of every droplet in the array varying randomly within a droplet diameter from the initial position. The resulting distribution of $S_{FS}$ values was used to obtain the mean feldspar-water overlapping area per droplet and its standard deviation, which was then used for calculation of the INAS

density. The values of $S_{FS}$ and corresponding standard deviations are given in the last column of Table 1. The standard deviations of $S_{FS}$ were used to estimate the uncertainty of the INAS density, which was found to be within ±100% from the mean value for all measurements conducted with the grain mounts.

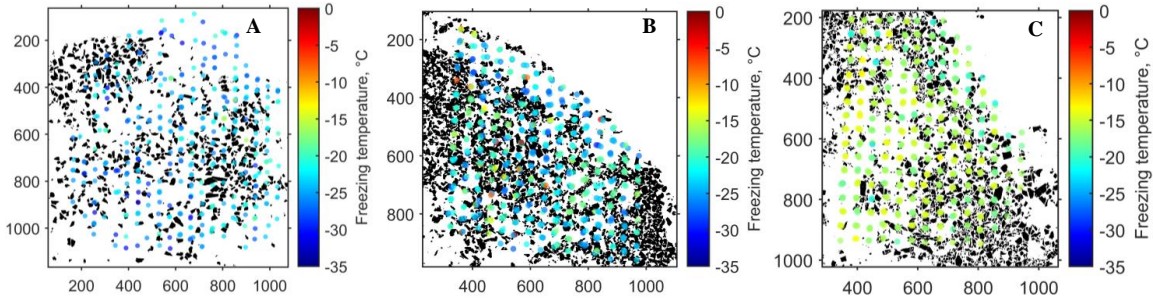

**Figure 3. Binary images of grain mounts (A) FS08-04, (B) FS0864O, and (C) FS0864C, overlaid with the heat maps of droplet arrays color-coded according to the droplet freezing temperatures. Heat maps for three repeated experiments are combined. The numbers on the X and Y axes are pixel coordinates of the images. The pixel size is 8.8 μm.**

For reference, the original Volkesfeld sanidine (sample FS08-VS) was prepared as a powder by grinding the specimen in a mortar and sieving the powder through a 20 μm sieve. The powder was then used for preparing an aqueous suspension

containing 0.1 wt% and 1 wt% feldspar. Suspension droplets with a volume of 21.6 nL were deposited in a 9 x 9 array with a PipeJet nano dispenser (BioFluidix GmbH) on a clean 10 x 10 mm silicon wafer (Ted Pella, Inc., product number 16006). About 10 droplets were deposited on top of the thin-film temperature sensor and could not be used for freezing measurements, so that approximately 70 droplets in a single experiment could be used. Freezing of the suspension droplets was measured using the same device as described above, see also Peckhaus et al. (2016). For the calculation of the INAS density of the

sample FS08-VS the specific surface area (SSA) of the feldspar powder was determined using an Autosorb iQ model 7 gas sorption system (Anton Paar 3P-Instruments, former Quantachrome Instruments). Using Ar as a sorbent gas at 87 K and applying Brunauer–Emmett–Teller (BET) theory (Brunauer et al., 1938), the SSA of sample FS08-VS was found to be



(1.8±0.2) m²/g, implying a total surface area of the feldspar particles contained in a single droplet of $(3.9 \pm 0.4) \times 10^{-9}$ m² for the 0.1 wt% suspension and of $(3.9 \pm 0.4) \times 10^{-8}$ m² for the 1 wt% suspension.

For the droplet freezing experiments on the thin section of Pakistan microcline (sample FS06-010), droplets with a volume of 1.4 nL were used. Accordingly, larger droplet footprint areas were taken into account when calculating the INAS density.

To estimate the possible effect of the purity of the water used in the experiments we conducted several freezing experiments with pure water droplets deposited on a clean silicon wafer. The silicon wafer was shown previously to have no effect on ice nucleation (Peckhaus et al., 2016), and the freezing of droplets occurs within a narrow temperature interval between -35°C

and -36°C as shown by the dashed curve in Figure 4. When freezing occurs at temperature higher than this, the presence of an ice nucleating substrate or of ice nucleating particles suspended in a water droplet is inferred.

**Table 1. Overview of the feldspar samples and experimental conditions**

| Sample label | Sample origin and preparation | Preparation for cold stage experiments | | Number of droplets | Droplet footprint area, $S_d$ [m²] | Contact area of feldspar-droplet interface, [m²] |
|---|---|---|---|---|---|---|
| FS08-VS | Sanidine from Volkesfeld, Eifel, Germany; grinded and sieved to < 20μm | Aqueous particle suspension | 0.1 wt% | 160 | 2.1×10⁻⁷ | (3.9±0.4)×10⁻⁹ |
| | | | 1 wt% | | | (3.9±0.4)×10⁻⁸ |
| FS08-Epx | Feldspar-free epoxy resin | Grain-free epoxy resin mount, ground and polished < 1 μm, ion-milled | | 540 | 1.2×10⁻⁸ | NA |
| FS08-01 | Albite-shifted sanidine, 8 days at 850°C in NaCl-KCl salt mixture melt | Grain mount, ground and polished < 1 μm, ion-milled | | 386 | 9.9×10⁻⁹ | (12.9±1.5)×10⁻¹⁰ |
| FS08-64o | Albite-shifted sanidine tempered at 550°C for 64 days in NaCl-KCl salt mixture melt | Grain mount, ground and polished < 1 μm, ion-milled | | 504 | 1.2×10⁻⁸ | (39.4±1.8)×10⁻¹⁰ |
| FS08-64c | Albite-shifted sanidine tempered at 550°C for 64 days in vacuum | Grain mount, ground and polished < 1 μm, ion-milled | | 523 | 7.8×10⁻⁹ | (16.7±1.2)×10⁻¹⁰ |
| FS06-010 | Pakistan feldspar (41% or, 39% mic, 20% ab), showing polysynthetic twinning according to the pericline law and K-Na exsolution lamella structure | Thin section for optical microscopy (~ 20 μm), polished to < 1 μm, ion milled | | 340 | 1.9×10⁻⁸ | 1.9×10⁻⁸ |

### 3 Results and discussion

**3.1 Droplet freezing experiments**

The ability of feldspar to induce freezing of supercooled water was investigated in a series of droplet freezing experiments as described in section 2.2.1. The experimental results are summarized in Figure 4. Panel A shows the fraction of frozen droplets as a function of supercooling temperature, and panel B shows the INAS density $n_s(T)$ calculated from the data presented in panel A and accounting for the contact area between supercooled water and feldspar (see Table 1). Representation of the results

of the freezing experiments in terms of $n_s(T)$ allows for comparison of the IN efficacy of various substrates and powder samples even if obtained by different methods.

The droplet freezing temperature is a function of substrate activity, droplet footprint area, and cooling rate, and therefore the freezing behaviour of different samples cannot be directly compared based on the freezing curves alone (Figure 4A). Nevertheless, some general features are clearly recognized. Freezing of more than 90% of pure water droplets on a Si wafer

occurs within a narrow temperature interval of 1 K. This behaviour is characteristic for homogeneous freezing of supercooled





water (see e.g. Ickes et al. (2015). The freezing curve of supercooled droplets on the epoxy resin (FS08-Epx) has a pronounced steep section between –28°C and –30°C, where the majority of the droplets freeze almost simultaneously. This freezing behaviour suggests that a single type of ice nucleating active sites, albeit with low freezing efficacy, prevails (Vali, 2008; Wright and Petters, 2013). The freezing curves of the samples containing feldspar exhibit less steep slopes pointing to an

intrinsic variability of ice nucleating active sites in contact with a droplet of supercooled water.

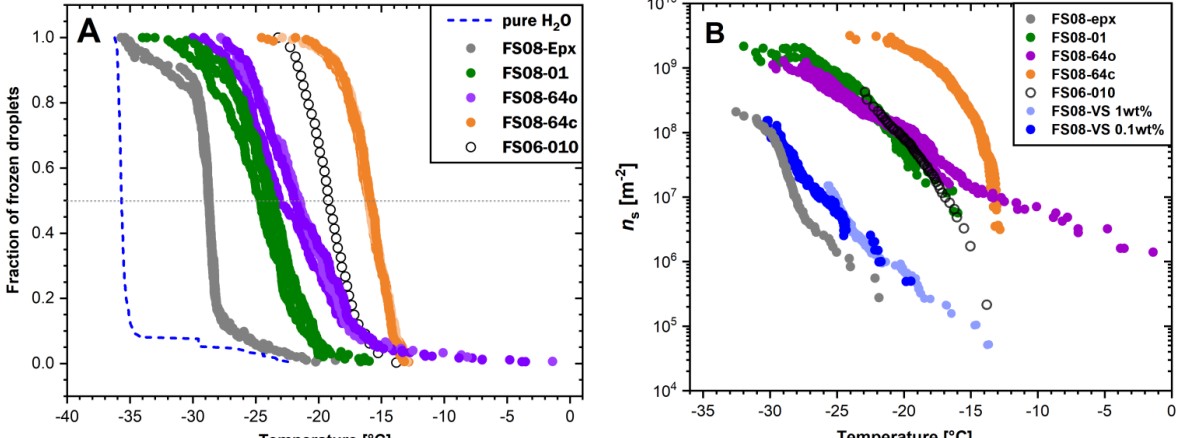

**Figure 4. Results of the droplet freezing experiments. A. Fraction of frozen water droplets as a function of supercooling temperature. The cooling rate was 3 K/min in all experiments. Pure water droplets on a clean silicon wafer freeze within a small temperature**
**range between -35°C and -36°C (as shown by the blue dashed curve). B. INAS density calculated from the data shown in panel A. In addition, the INAS density calculated for the untreated Volkesfeld sanidine (sample FS08-VS) obtained from suspension droplet freezing experiments is shown for comparison in two different shades of blue for the two weight concentrations (dark blue for 0.1 wt% and light blue for 1 wt%).**

Figure 4, panel B, shows the $n_s(T)$ curves obtained from the chemically altered feldspar and from the original Volkesfeld
sanidine, together with the $n_s(T)$ curves for the epoxy resin and of the microcline from Pakistan (FS06-010). As expected, the epoxy resin has the lowest INAS density, followed by the original feldspar FS08-VS, the feldspar FS08-01 that was cation-exchanged at 850°C, and the feldspars, which were first cation-exchanged at 850°C and then annealed at 550° in an open system setting (sample FS08-64o) and in a closed system setting (sample FS08-64c). Note that the original Volkesfeld sanidine was prepared as a powder and suspension droplets were used for the freezing experiments. Grinding of the material may
introduce morphology changes beyond a general increase of the specific surface area (Hiranuma et al., 2014), and the comparison of its IN activity with the IN activities of the other samples, which were determined from grain mounts, needs to be considered with caution. Despite of this uncertainty, it may be stated that the INAS density appears to increase substantially after the ion-exchange treatment. All ion-exchanged feldspars have substantially higher INAS densities than the untreated Volkesfeld sanidine. Recalling that Volkesfeld sanidine was used as the starting material for the cation exchange experiments,
the observed difference in INAS density is ascribed to the effects of cation exchange. Apart from a shift in chemical composition, the most obvious effect of cation exchange is the generation of a system of parallel cracks that are oriented sub-parallel to the Murchison plane. The Murchison plane encloses an angle of about 8° to 11° with the (100) lattice plane and, given that the crack flanks are somewhat uneven, they probably contain patches of (100) crystal surface. An increase of the INAS density due to exposure of (100) crystal surfaces is well in line with the findings of Kiselev et al (2017), who observed
preferential epitaxial nucleation and growth of ice crystals on (100) surfaces of alkali feldspar. It was shown by the latter authors that the atomic structures of the (10-10) prism planes of ice and the (100) planes of alkali feldspar have striking similarities allowing for efficient epitaxial nucleation and growth of ice on (100) surfaces of alkali feldspar. Interestingly, the $n_s(T)$ of the cation exchanged feldspar FS08-01 is very similar to the INAS density of the (010) thin section of Pakistan





feldspar (sample FS06-010). The comparison between the two feldspar types is perfectly justified as identical preparation and

measurement routines were applied for both samples.

The annealing of the cation exchanged samples at 550°C lead to a further increase of the INAS density. Both, sample FS0864o, which was annealed in contact with the salt – open system setting, and sample FS0864c, which was annealed in vacuum – closed system setting, show higher INAS densities than sample FS08-01, which was prepared from cation exchanged Volkesfeld sanidine without subsequent annealing. Samples FS08-64o and FS08-64c show quite different freezing behavior.

Whereas sample FS08-64c has a steep temperature dependency and the highest $n_s(T)$ values of all samples, the $n_s(T)$ curve of sample FS08-64o is less steep and extends to comparatively high temperatures indicating that a set of very active ice nucleating sites is capable of triggering ice nucleation at temperatures as high as -3°C. However, at temperature below -15°C, the FS08-64o sample has the same INAS density as the FS08-01. Ice nucleation at similarly high temperatures was reported from sample FS04 studied in Peckhaus et al. (2016).

**3.2 Grain morphology and chemical composition**

The grain morphologies and chemical compositions were characterized by ESEM for all samples except FS06-010 which was additionally studied with powder XRD (see Table 1). ESEM images of the cation exchanged sample FS08-01 and of the cation exchanged and annealed samples FS0864o and FS0864c are shown in Figure 5 and in supplementary Figure 3. The most evident feature of all samples is the presence of a system of parallel cracks. The cracks extend at high angles to the (010) and

the (001) cleavage planes bounding the feldspar fragments. In the cation exchanged feldspar (FS08-01) the cracks are, however, hardly visible, whereas they are up to 5 μm wide and well visible in the annealed feldspars. In the grains of sample FS0864o, K-rich zones have developed along the cracks. In the back-scattered electron (BSE) image the K-rich zones appear as light grey bands flanking the cracks on both sides (dashed arrows in Figure 5B). The K-rich zones were formed by successive Na-K exchange between the feldspar and the salt during annealing at 550°. At this temperature potassium partitions more strongly

into the feldspar than during the primary cation exchange at 850°C. This induced back in-diffusion of K during annealing at 550°C, which locally reversed the compositional shift attained during the primary cation exchange at 850°C. No such K-rich zones are visible in the BSE images of feldspar FS0864c. This is due to the fact that this sample was separated from the salt after primary cation exchange and subsequently annealed in vacuum at 550°C. The feldspar fragments of this sample show uniform BSE contrast and appear to be compositionally homogeneous. From recent work (Petrishcheva et al., 2020) it is

known, however, that during the annealing at 550°C an alkali feldspar with $c_K = 0.41$ experiences phase separation by spinodal decomposition. At the applied experimental conditions and annealing time the spinodal decomposition typically produces an alternation of more Na-rich and more K-rich lamellae with a characteristic lamellar spacing on the order of 30 nm (Petrishcheva et al., 2020), a microstructure that is referred to as cryptoperthite (Smith and Brown, 1988). The nm-scaled lamellar intergrowth cannot be resolved with BSE imaging in the SEM and a uniform grey shade is observed. It has been shown by Petrishcheva et

al. (2020) using Scanning Transmission Electron Microscopy (STEM) that the lamellae are coherently intergrown so that lamella boundaries are free of misfit dislocations and the lattice misfit between the more Na-rich and the more K-rich lamellae is compensated by elastic strain. In contrast, misfit dislocations are frequently observed at lamellar boundaries in natural perthites (Lee et al., 1995; Abart et al., 2009).

The general enhancement of the IN activity by the chemically induced cracks is corroborated by the presence of abundant

parallel cracks in all cation exchanged samples. With respect to the differences in the INAS density between the cation exchanged sample FS08-01, and the cation exchanged and subsequently annealed samples FS0864c and FS0864o it may be speculated that the narrow cracks of sample FS08-01 were less accessible to water than the more open cracks of the annealed samples. The fact that the $n_s(T)$ curve of sample FS0864c has quite similar shape as the $n_s(T)$ curve of sample FS08-01 and is only shifted to higher temperatures suggests that annealing of sample FS0864c in vacuum only lead to a widening of the

cracks but did not change their morphological characteristics. In contrast, the characteristics of the INAS was substantially




modified during the annealing of sample FS0864o in contact with salt. It is known from earlier work (Neusser et al., 2012; Schäffer et al., 2014a) that lattice expansion associated with a compositional shift of Na-rich alkali feldspar towards more K-rich compositions may lead to the formation of cracks subparallel to the grain surface and disintegration of the crystal by spalling. Formation of the K-rich zones flanking the cracks by in-diffusion of K from the crack surfaces into the cation

exchanged felspar during annealing at 550°C in contact with the salt may thus have produced secondary cracks approximately parallel to the crack walls that had previously been induced during primary cation exchange. These secondary cracks may explain the second population of INAS inferred from the $n_s(T)$ curve of sample FS0864o. The mechanisms underlying chemically induced fracturing are addressed in the next section.

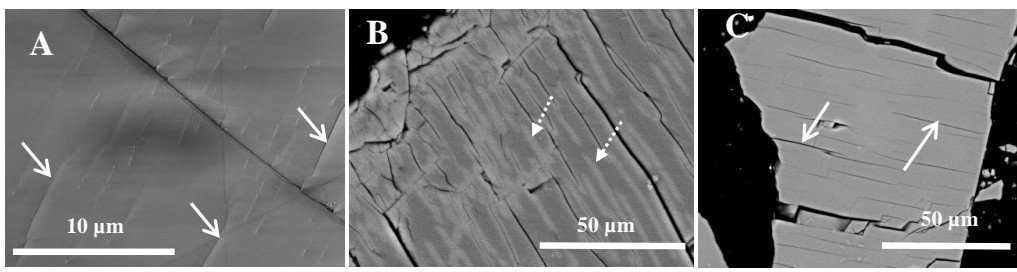


**Figure 5. Back-scattered electron (BSE) images of feldspar grains at different stages of the treatment. The arrows mark chemically induced cracks extending approximately parallel to the Murchison plane. A. Surface of a sanidine grain after a compositional shift from cK = 0.84 to cK = 0.43 by cation exchange (cation exchanged sample FS08-01). B. Surface of a cation exchanged sample after annealing at 550°C for 64 days in contact with salt (open system setting, sample FS0864o). The light grey bands (marked by the**

**dashed arrows) are K-rich zones that developed by reverse cation exchange along the cracks, dark areas correspond to the regions with higher concentration of Na. C. Surface of cation exchanged sample after annealing at 550°C in vacuum for 64 days (closed system setting, sample FS0864c). At the resolution of ESEM this sample appears chemically homogeneous despite of the fact that the sample is exsolved forming a lamellar intergrowth with characteristic lamellar spacing of about 30 nm (Petrishcheva et al., 2020). The relatively straight grain edge on the right follows a (010) cleavage plane.**

**3.3 Chemically induced fracturing in alkali feldspar**

Generally, the lattice parameters of a crystalline solid-solution depend on composition, and mechanical stress may be induced in a single crystal, when it undergoes inhomogeneous compositional change (Larche and Cahn, 1982). If in a brittle crystal the stress level exceeds a critical value, it will undergo fracturing. The rationale underlying the exploitation of this phenomenon for generating cracks in alkali feldspar is based on the notion that alkali feldspar is a brittle mineral and its lattice parameters

show a strongly anisotropic compositional dependence. Generally, the crystal structure of alkali feldspar contracts with a compositional shift from K-rich towards more Na-rich compositions, where the effect is largest in the direction of the crystallographic a-axis and less pronounced parallel to the crystallographic b- and c-axis (Kroll et al., 1986; Angel et al., 2012). It is known from earlier work (Petrishcheva et al., 2014; Petrovic, 1972; Schäffer et al., 2014b) that the rate of Na-K exchange between alkali feldspar and an NaCl-KCl salt melt is controlled by the interdiffusion of Na and K in the alkali feldspar. The

in-diffusion of Na into a single crystal of K-rich alkali feldspar thus produces a Na-rich surface layer the thickness of which increases with time.

According to the compositional dependence of the lattice parameters of alkali feldspar, a compositional eigenstrain is induced within the Na-rich surface layer, which is characterized by contraction primarily sub-parallel to the crystallographic a-direction. The Na-rich surface layer and the unaltered alkali feldspar beyond the chemically altered surface layer pertain to a

single solid and are mechanically coupled, and, as long as the chemically altered surface layer is thin compared to the size of the unaltered core region of the grain, the chemically induced lattice contraction in the Na-rich surface layer is largely compensated by elastic distension to keep the lattice dimensions compatible across the compositional transition zone separating the unaltered core from the chemically altered surface layer. This leads to a tensile stress state in the Na-rich surface layer, where the maximum tensile stress component is approximately parallel to the direction of maximum lattice contraction. If this



tensile stress exceeds about 300 MPa, the surface layer yields by fracturing (Neusser et al., 2012; Scheidl et al., 2014) producing a set of parallel cracks oriented approximately perpendicular to the a-direction, which is close to the orientation of the Murchison plane and of the (100) lattice plane. The cracks show quite uniform spacing, which depends on the extent of the applied compositional shift (Scheidl et al., 2014). As soon as the cracks are formed, they are infiltrated by the salt melt, and the crack walls serve as new surfaces, where cation exchange takes place. This leads to a situation, where the cracks propagate

independently of the stress state at the original crystal surface leading to complete disintegration of the original grain into much smaller fragments (Petrishcheva et al., 2019; Predan et al., 2020). As the fragments are in part bound by the crack surfaces, and these are sub-parallel to the (100) lattice plane, these particles have a high proportion of (100) surface exposed. In addition, larger particles have cracks with orientations close to (100). Although shifting of K-rich alkali felspar towards more Na-rich compositions by cation exchange is an efficient route for producing (100) crystal surfaces in experiment, this mechanism very

likely is not relevant in nature. Nevertheless, similarly oriented parting planes following the Murchison plane may be generated by eigenstrain effects during perthite formation via hydrothermal alteration. This mechanism is addressed in the next section.

**3.4 Formation of (100) feldspar surfaces by exsolution and their potential contribution to the enhancement of the IN efficacy**

Above about 600°C, depending on pressure and Ca content, alkali feldspar shows complete miscibility. Towards lower

temperature, a homogeneous solid-solution of intermediate composition is not thermodynamically stable (Brown and Parsons, 1984). If alkali feldspar of intermediate composition is cooled into the two-phase region of its phase diagram, it exsolves typically forming a lamellar intergrowth of more Na-rich and more K-rich domains giving raise to perthite microstructure (Brown and Parsons, 1984). The lamellar intergrowth is coherent, at least during the early stages of exsolution and due to the compositional dependence of the lattice parameters of alkali feldspar (Angel et al., 2012; Kroll et al., 1986) the compositionally

distinct lamellae exhibit considerable lattice misfit. In coherent intergrowth this lattice misfit is compensated by elastic deformation of the lamellae. The strong anisotropy of the chemically induced eigenstrain exerts a first-order control on the orientation of the exsolution lamellae. Typically, the exsolution lamellae are oriented parallel to the Murchison plane, which ensures the minimum possible crystallographic misfit between the Na-rich and the K-rich lamellae and is oriented approximately perpendicular to the direction of maximum compositionally induced eigenstrain (Laves, 1952; Robin, 1974;

Williame and Brown, 1974). Orientation of the exsolution lamellae parallel to the Murchison plane ensures that the elastic strain energy associated with coherent intergrowth of Na-rich and K-rich lamellae is minimized. Nevertheless, coherency stress is induced during exsolution. In natural exsolved alkali feldspars the lattice misfit at the lamella interfaces is partially accommodated by edge dislocations lying in the interface planes (Lee and Parsons, 1995; Fitz Gerald et al., 2006; Abart et al., 2009). Around these edge dislocations the feldspar is strained and prone to dissolution in the course of sub-solidus

hydrothermal alteration and weathering (Parsons et al., 2005; Lee et al., 1995). Parting planes following the Murchison pane are known from *murchisonite*, a variety of perthite characterized by pseudo cleavage along these non-rational planes (Bollmann and Nissen, 1968). At any rate, alignment of edge dislocations at the lamella interfaces leads to mechanical weakening of these interfaces and fosters disintegration of the feldspar by fracturing along the lamella interfaces. The lamellar interfaces and, hence, the related parting planes are sub-parallel to the (100) plane. Disintegration of exsolved perthitic alkali feldspar due to

mechanical stress thus tends to occur along the lamella interfaces and thus very likely exposes patches of (100) surfaces on the disintegrated feldspar particles (Parsons et al., 2005). In addition, the line defects along the semi-coherent lamellar boundaries may develop into corrosion channels during hydrothermal alteration and weathering (Parsons, 1978; Lee et al., 1995). Such corrosion channels may become sufficiently large to serve as nucleation sites for ice crystals on patches of (100) surfaces exposed on the walls of such corrosion channels. The fact that deposition ice nucleation on feldspar samples was often observed

to be associated with surface defects (Kiselev et al., 2017; Pach and Verdaguer, 2019), suggests that the ice active sites must be exposed to liquid water which is likely to form inside the cracks due to capillary condensation (Kanji et al., 2017; Koop,





2017; Marcolli, 2014; David et al., 2019). Finally, the elastic strain energy that is stored in coherent lamellar intergrowth in exsolved alkali feldspar increases its Gibbs energy and makes these minerals prone to fluid-mediated mineral replacement (Brown and Parsons, 1993; Parsons and Lee, 2008). Fluid-mediated mineral replacement of exsolved alkali feldspar produces

nano-porosity (Putnis, 2002; Walker et al., 1995; Worden et al., 1990), which is concentrated along lamellar interfaces (Abart et al., 2009; Tajčmanová et al., 2012) and likely exposes patches of (100) crystal surfaces, which may serve as ice nucleation sites. This scenario probably applies to the natural Pakistan feldspar (sample FS06-010), which has porosity potentially induced by hydrothermal processes or deuteric alteration.

### 4 Conclusions

One of the possible explanations suggested for the repeatedly observed ice nucleation in the pores and cracks on the surface of alkali feldspars is the presence of small patches of crystal surface with (100) orientation that are exposed in the cracks due to natural fracturing or hydrothermal/deuteric alteration of alkali feldspar. Up to now, experimental evidence corroborating this hypothesis was missing. Here, we test this hypothesis experimentally and propose mechanism explaining why such (100) surfaces are preferentially found in cracks in alkali feldspar. To this end, K-rich gem quality alkali feldspar was shifted towards

more Na-rich compositions by cation exchange with molten NaCl-KCl salt at 850°C and ambient pressure to exploit the associated anisotropic chemically induced contraction of the crystal structure. Through this treatment, a system of parallel cracks with orientations approximately parallel to the Murchison plane, an irrational plane in alkali felspar along which the lattice misfit between more Na-rich and more K-rich alkali feldspars is minimized, was generated. The Murchison plane is oriented close to (100) and the somewhat uneven crack surfaces likely contain patches of (100) crystal surfaces. A substantial

enhancement of the overall INAS density in the cation exchanged samples as compared to the untreated reference material corroborates the high IN activity of the chemically induced cracks, which we relate to the presence of patches of (100) crystal surfaces on the crack surfaces. Annealing of the cation-exchanged alkali feldspars at 550°C in vacuum subsequently to the primary cation exchange lead to a widening of the cracks, which further enhanced the ice nucleating efficacy of the sample. Annealing of the cation-exchanged feldspar at 550°C in the presence of the salt, lead to the formation of a K-rich surface layer

along the crystal surfaces and along the surfaces of the previously induced cracks due to reversed cation exchange at the lower annealing temperature. This supposedly induced secondary, surface parallel cracks, which further enhanced the IN activity and created new types of INAS with IN activity at temperatures as high as -3°C. Our results confirm that chemically induced fracturing in alkali feldspar is a viable mechanism for increasing the INAS density of alkali feldspar. This mechanism is, however, unlikely to play a significant role in the conditioning of natural feldspar, as interaction between alkali feldspar with

an inorganic salt melt is unlikely in natural environments. In natural alkali feldspars, parting planes following the Murchison plane may, however, occur due to the mechanical effects associated with exsolution. Separation of initially homogeneous alkali feldspar into lamellae of more Na-rich and a more K-rich alkali feldspar during cooling is a widespread phenomenon in natural alkali feldspar. The chemical eigenstrain associated with exsolution and the resulting lattice mismatch between the two phases is accommodated by elastic strain and/or by the introduction of misfit dislocations at the lamellar boundaries. Both phenomena

make lamella interfaces prone to alteration and corrosion during hydrothermal overprint or weathering, which may eventually lead to parting along the Murchison plane. The evolution of a perthite microstructure in alkali feldspar thus fosters the exposure of (100) crystal surfaces and thus enhances the IN activity of exsolved alkali feldspars, a scenario which most likely applies to sample FS06-010 investigated in this study.

### Data availability

The data represented in Figure 3 and Figure 4 has been made publicly available in KITOpen Repository under doi.org/10.5445/IR/xxxxxxxxx.



## Author contributions

AAK and RA conceived the idea of the study and wrote this manuscript with contributions from all authors. AK and TG
conducted the droplet freezing assay experiment and evaluated the results with contribution from TL and AAK. CS and EP
conducted cation exchange experiments and sample preparation. All authors contributed to the discussion of the results.

## Competing interests

The authors declare that they have no conflict of interest.

## Acknowledgements

The authors acknowledge financial support of German Research Foundation (DFG) under Grant KI 1997/1-1 and from the
Austrian Science foundation FWF Grant I 4404-N. as well as support by the Helmholtz Association under Atmosphere and
Climate Programme (ATMO). The authors also greatly acknowledge XRD analysis of FS06 sample by Dr. Jörg Göttlicher
(KIT, IPS) as well as initial cold stage measurements conducted by Mr. Sören Bergmann.

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
