# Peer review of "Effect of chemically induced fracturing on the ice nucleation activity of alkali feldspar."

_Atmospheric Chemistry and Physics, 2021_

## Author Comment (AC1)

**Response to reviewer comments for ACP-2021-18 (Kiselev et al.)**

**General**

We would like to thank Dr. Whale and the anonymous reviewer for their encouraging and thoughtful comments that helped us to improve the manuscript. We have now revised our manuscript accordingly.

Below, we respond (green) to each individual comment. Where appropriate, we also show the changes that were made to the manuscript (blue).

RC1: 'Review of Kiselev et al.', Thomas F. Whale, 11 Mar 2021

This paper demonstrates that lamellar microstructures on feldspar mineral surfaces are at least partially responsible for the exceptional ice nucleating ability of some feldspars, confirming the suggestion in previous work that such structures might be partially causative of the observed ice nucleating effectiveness. This was done by inducing formation of the microstructures on a previously microstructurally pristine feldspar through high temperature chemical treatment, then demonstrating the treated feldspars nucleate ice far more effectively than the untreated sample from which they were produced. While the method used to produce the modified feldspar is not directly relevant to feldspar that might end up in the atmosphere the authors argue convincingly that ubiquitous perthitic feldspars will likely possess similar surface features. It is a striking result and really excellent work, in my view. I particularly like the solution to the challenge of investigating the ice nucleation ability of the small quantities of altered feldspar produced. The paper is well written and clear, represents an important contribution to understanding of how natural mineral dusts might interact with clouds and seems to me entirely suitable for publication. I have a few minor comments and suggestions that should be addressed.

**Minor comments**

There a couple of places in the paper that suggest exposure of the (100) face of feldspar by surface cracks is more established than I understand it to be. For the most part, the hypothesis that the feldspar (100) face is responsible for ice nucleation is appropriately presented as a hypothesis. However, on line 54 Fitz Gerald et al. (2006) is cited as saying ...patches of (100) crystal surfaces may be exposed at defects such as cracks and cavities.' I do not think Fitz Gerald et al. (2006) says this, only that there are various kinds of nanoscale features in microtexturally complex feldspar. While Kiselev et al. (2017) argues that patches of the (100) face could be responsible for observed ice nucleation and Whale et al. (2017) agrees that the (100) face could be responsible neither demonstrates that there are indeed (100) faces present in the defects discussed. Similarly, on line 395-396 Parsons et al. (2005) doesn't say anything about the (100) plane as far as I can see. It is perhaps a minor point, and I quite agree that it is likely that (100) face is exposed in these cracks, but I think it might be best to be really clear about this, lest it becomes common knowledge that cracks on feldspar expose the (100) face when, to the best of my knowledge, this has not been conclusively demonstrated. There are authors on the paper who are clearly much more knowledgeable than me as far as mineral structures go, and I may have misunderstood what is said in the cited papers, however I would like this clarified.

We agree with the reviewer that the exposure of the (100) face in the defects was not mentioned explicitly in the works of Fitz Gerald et al. (2006) and Parsons et al., (2005). Both works, however, explicitly mention the Murchison parting plane being the plane where the majority of strain-controlled dislocations are present. As explained in the manuscript, Murchison plane, oriented between (-801) and (-601), is subparallel to (100) within just few degrees. Being an irrational plane, Murchison plane cannot be exposed as an atomically flat surface and must include faces of rational planes, of which the (100) is the most closely oriented. It is thus very likely that patches with (100) orientation are exposed in the cracks with the general orientation defined by the Murchison plane.

We have removed the ambiguous citations and modified the text of the manuscript to make it clear. We have also prepared a sketch illustrating the orientation of Murchison plane relative to the main crystal planes of feldspar (see Supplementary material).

I'm a little surprised Holden et al. (2019) hasn't been cited or discussed. This work shows that ice invariably nucleates on the micron-size surface pits prevalent on natural feldspars surfaces in the immersion mode. This observation is mentioned in the paper (notably at the start of the conclusions) but isn't cited. The other papers cited for this looked at nucleation of ice from vapour rather than from liquid water, as is investigated in this paper.

Holden et al. (2019) is now cited in the introduction and in the discussion section 3.4. For the sake of completeness, we think that the difference between ice nucleation in immersion and deposition mode is not fundamental, especially in light of the recent theoretical and experimental studies of pore condensation and freezing (PCF) (see, for example, David et al., 2019)

For completeness, I would prefer if the paper also noted that molecular dynamics simulations have not found preferential ice nucleation on the (100) face of feldspar (Soni and Patey, 2019). I don't think it would add many words and would give an essentially complete picture of where this area of study is at the moment, so it seems to me appropriate to mention this paper.

We agree with Dr. Whale that the citation of this work (Soni and Patey, 2019) makes the overview of the current research more complete. We thus add a sentence summarizing the main findings of their work in the introduction and further mention their study in the discussion section.

It might be worth spelling out what 'sub-parallel' means somewhere. Much of the readership of this work may not be that familiar with mineralogical terminology. Similarly, there are other words that may benefit from a quick description. 'Spalling' on line 323 for instance.

In the introduction, we explain that the Murchison plane is oriented between (-601) and (-801) and thus the angle between the Murchison plane and (100) is between 8° and 11°. We now explicitly use the term "sub-parallel" in this sentence to avoid misunderstanding.

Line 267- I would suggest 'in spite of' rather than 'despite of'

OK

Line 279-280. It doesn't seem correct to say that 'identical preparation and measurement routines were applied for both samples' to me. I agree a comparison is reasonable but the (010) FS06-010 thin section presumably presents only that crystal face, for the most part, where the grains embedded in epoxy presumably present a fairly random sampling of crystal faces?

The preparation and measurement routines were identical in the sense that the same workflow was used to prepare the thin section samples; the difference between the original samples (randomly oriented grains vs. single crystal section) is obvious. We have modified the wording of this sentence to clarify this issue.

Line 296- missing hyphen in FS0864o, there is some inconsistency with hyphen use in sample names elsewhere also.

**Corrected throughout the text**

Line 320- I would see widening as a change in morphological characteristic, and I am not really sure how either a widening or change in morphological character would be expected to impact on ice nucleation temperatures. Are the authors suggesting that wider cracks might expose more (100) face? It might help if this section is a little more specific.

We agree with Dr. Whale that "widening" is perhaps not the best term to describe the process of chemically induced propagation of cracks. The cracks become deeper and longer, thus exposing larger surface that might contain patches with (100) orientation.

Line 325- feldspar misspelt

corrected

Line 426- I'm not sure 'supposedly' is the right word?

definitely not the right word, corrected

**References**

David, R. O., Marcolli, C., Fahrni, J., Qiu, Y., Perez Sirkin, Y. A., Molinero, V., Mahrt, F., Brühwiler, D., Lohmann, U., and Kanji, Z. A.: Pore condensation and freezing is responsible for ice formation below water saturation for porous particles, Proceedings of the National Academy of Sciences, 116, 8184-8189, 10.1073/pnas.1813647116, 2019.

Fitz Gerald, J. D., Parsons, I., and Cayzer, N.: Nanotunnels and pull-aparts: Defects of exsolution lamellae in alkali feldspars, American Mineralogist, 91, 772-783, doi:10.2138/am.2006.2029, 2006.

Holden, M. A., Whale, T. F., Tarn, M. D., O'Sullivan, D., Walshaw, R. D., Murray, B. J., Meldrum, F. C., and Christenson, H. K.: High-speed imaging of ice nucleation in water proves the existence of active sites, Science Advances, 5, eaav4316, 10.1126/sciadv.aav4316 %J Science Advances, 2019.

Kiselev, A., Bachmann, F., Pedevilla, P., Cox, S. J., Michaelides, A., Gerthsen, D., and Leisner, T.: Active sites in heterogeneous ice nucleation—the example of K-rich feldspars, Science, 355, 367-371, 10.1126/science.aai8034, 2017.

Parsons, I., Thompson, P., Lee, M. R., and Cayzer, N.: Alkali Feldspar Microtextures as Provenance Indicators in Siliciclastic Rocks and Their Role in Feldspar Dissolution During Transport and Diagenesis, Journal of Sedimentary Research, 75, 921-942, 10.2110/jsr.2005.071, 2005.

Soni, A., and Patey, G. N.: Simulations of water structure and the possibility of ice nucleation on selected crystal planes of K-feldspar, 150, 214501, 10.1063/1.5094645, 2019.

Whale, T. F., Holden, M. A., Kulak, A. N., Kim, Y.-Y., Meldrum, F. C., Christenson, H. K., and Murray, B. J.: The role of phase separation and related topography in the exceptional icenucleating ability of alkali feldspars, Phys. Chem. Chem. Phys., 19, 31186-31193, 10.1039/C7CP04898J, 2017.

**RC2: 'Comment on acp-2021-18', Anonymous Referee #2, 12 Mar 2021 reply**

**General Comment**

This manuscript focuses on identifying the potential role of (100) plane/close to (100) orientation in the exceptionally high ice nucleating ability of perthitic alkali feldspar. This has been achieved using a combination of carefully designed experimental procedure and laboratory-based techniques which complement each other. The authors make a great use of the knowledge/literature on feldspar mineralogy in deducing factors affecting the ice nucleating ability of the mineral class. While I fully support the publication, I do have few minor remarks that the authors should address while preparing the final version of the manuscript.

**Minor comments:**

Authors are encouraged to incorporate a sketch in either the Introduction or Sect 2.1.1 that shows/highlights the feature of Murchison plane in reference to a standard crystal lattice for an easy visual understanding. (Lines 58-61)

We have incorporated a sketch illustrating orientation of the cracks and of the Murchison plane relative to the (100) plane and placed it into the supplementary material (Supplementary Figure 4). The figure is cited in the Introduction.

Did the authors observe any perceivable changes in the structure of the Na-rich, chemically induced crack regions after one or several freezing events? Given that the frozen fraction curves do not differ much over replicate measurements on a grain mount (e.g. FS08-64o), is it safe to assume that the morphology of such structures are quite stable? (in reference to Figure 3 & 4)

We have not conducted comparative morphology study before and after the ice nucleation measurements. As correctly noticed by the reviewer, the good repeatability of the replicate measurements indicates a stability of the morphological features responsible for ice nucleation behavior.

Line 146 It is not very clear why authors specifically chose to test the (010) plane (and not as powder suspension) of alkali feldspar from Pakistan (sample FS06-010). A short explanation can be added regarding this.

The study of (010) is a part of a larger study comparing nucleation on various faces. The manuscript describing the results of this study is under preparation. We mention it at the end of the sample preparation section of the manuscript.

A brief yet critical discussion on the drastic difference between the ice nucleating ability of the chemically modified samples (grain mounts; FS08-64o/64c/01) and the original powder (FS08-VS) is missing. Does the latter have entirely different nature of ice active sites? Or do the authors suggest that the ice nucleating ability in powder form still originates from the perthitic structures, albeit further enhanced after chemical modification?

The original sanidine (sample FS08-VS) is a gem quality sanidine, which is homogenous on the nm scale. It is devoid of cracks inclusions or any other flaws, nor does it show perthitic structure. Our interpretation is that random milling of a single crystal of this material produces fragments with mostly (001) and (010) cleavage panes forming the surfaces. These particles very likely only expose a subordinate fraction of (100) oriented patches. We added this interpretation to the conclusion section.

Line 52-53 Is that a common feature amongst all alkali feldspars or authors are referring to any specific type? Needs clarification

We have modified the wording of these sentences to make the explanation clearer:

Facets with (100) orientation, however, do not pertain to the commonly exposed crystal surfaces of alkali feldspar. The surfaces of natural feldspar are either represented by the typical growth facets including the (110), (-101), (001), (20-1) and (010) facets (Smith and Brown, 1988, or by the (001) and (010) cleavage planes, where the cleavage is perfect on the (001)- and good on the (010) plane.

Lines 92-93 How did the authors determine whether the surface attained equilibrium with the solution? Did they monitor the change in composition over time? Needs clarification

We agree with the reviewer that this point should be explained clearly. At 850°C Na-K exchange equilibrium between the surface of alkali feldspar and an NaCL-KCI salt melt is closely approached (to within about 1 mole %) within a few hours. The chemically altered surface layer grows in thickness with time. The growth in thickness of the chemically altered surface layer is controlled by the interdiffusion of Na and K on the alkali sublattice of the feldspar. According to the calibration of Na-K interdiffusion in alkali feldspar by Schaeffer et al (2014) and by Petrishcheva et al (2014, 2020) the diffusion front would propagate about 10 to 20  $\mu$ m, depending on orientation in 8 days. As shown by Petrishcheva et al (2019) fracturing is already induced after about 2 days, and the crack flanks serve as new surfaces for cation exchange.

We added this paragraph to the section 2.1.1 describing the cation exchange and annealing experiments.

Lines 93-94 As mentioned earlier, this compositional alteration happens only at the surface. Can the authors comment on the depth and fraction of surface area altered (either qualitative or quantitative), if possible?

Please see our answer to the previous comment.

Lines 115-116 Was the third batch of particles exposed to lab/surrounding conditions during this annealing process to room temperature? This should be clearly stated

A third batch of cation exchanged feldspar grains was removed from the melt after the primary cation exchange. It was rinsed with distilled water to remove the salt and then gently dried and stored without further treatment.

Line 275-278 In reference to Kiselev et al. (2017), can the authors comment on the findings of Soni and Patey (2019) regarding (100) surface. A brief discussion on this would be useful as the current manuscript builds upon the previous findings

We now mention the negative results of the MD simulation study by Soni and Patey (2019) in the manuscript. However, we refrain from going into discussion of molecular nucleation mechanisms on various crystalline planes as being outside of the focus of this paper.

Line 300 "...at 850 °C." reference needed

This is an observation. The text was changed accordingly to make this clear!

Line 341 "...depend on composition,.." composition of what?

Generally, the lattice parameters of a crystalline solid-solution depend on its chemical composition. That has been clarifies in the revised manuscript.

**Technical comments:**

Line 25 ... Earth's...

**corrected**

Line 34 Can add abbreviation 'IN' here as 'ice nucleation' appears here the first time in the main text

**agreed, corrected**

Line 51 I assume the authors mean  $\overline{1}$ , as the notation of the planes (not (-101) and (20-1)). The notation needs to be corrected throughout the manuscript

corrected throughout the manuscript

Line 86 ....an NaCl-KCl....

corrected

Figure 1: Tick marks missing on X-Y axes

corrected

Line 205 unusually large spacing at ".. of SFS.."

**corrected**

Figure 4: There are unidentified data sets in beige and violet color (underlying FS08-64c & 64o) in Panel A. Please check

yes, color shading was used to indicate different runs of the freezing experiment, is now made consistent

Line 301 "...550°C locally reverses the ... "

Line 335 "...that are developed..."

Line 375 "thermodynamically unstable" instead of "not thermodynamically stable"

Line 377 ".....giving rise to..."

Line 435 "overprint or" not needed

Line 451 delete period sign after "...4404-N"

all of the above, corrected as suggested

**References**

Kiselev, A., Bachmann, F., Pedevilla, P., Cox, S. J., Michaelides, A., Gerthsen, D., and Leisner, T.: Active sites in heterogeneous ice nucleation—the example of K-rich feldspars, Science, 355, 367-371, 10.1126/science.aai8034, 2017.

Soni, A., and Patey, G. N.: Simulations of water structure and the possibility of ice nucleation on selected crystal planes of K-feldspar, J. Chem. Phys., 150, 214501, 10.1063/1.5094645, 2019